# Generative Table Pre-training Empowers Models for Tabular Prediction

**Tianping Zhang[1], Shaowen Wang[2], Shuicheng Yan[3], Jian Li[*1], Qian Liu[*4]**

[1]Tsinghua University, [2]Fudan University, [3]Skywork AI, [4]Sea AI Lab

{ztphs980, shuicheng.yan}@gmail.com, wangsw19@fudan.edu.cn,

liuqian@sea.com, lijian83@mail.tsinghua.edu.cn

## Abstract

Recently, the topic of table pre-training has attracted considerable research interest. However, how to employ table pre-training to boost the performance of tabular prediction remains an open challenge. In this paper, we propose TAPTAP, the first attempt that leverages table pre-training to empower models for tabular prediction. After pre-training on a large corpus of real-world tabular data, TAPTAP can generate high-quality synthetic tables to support various applications on tabular data, including privacy protection, low resource regime, missing value imputation, and imbalanced classification. Extensive experiments on 12 datasets demonstrate that TAPTAP outperforms a total of 16 baselines in different scenarios. Meanwhile, it can be easily combined with various backbone models, including LightGBM, Multilayer Perceptron (MLP) and Transformer. Moreover, with the aid of table pre-training, models trained using synthetic data generated by TAPTAP can even compete with models using the original dataset on half of the experimental datasets, marking a milestone in the development of synthetic tabular data generation. The code and datasets are available at `https://github.com/ZhangTP1996/TapTap`.

## 1 Introduction

Recently, pre-trained language models (LMs) have attracted a lot of research interest in different domains, especially in the area of natural language processing. After pre-training on a large-scale unstructured text corpus with a self-supervised training objective, e.g., masked language modeling (MLM) proposed by BERT (Devlin et al., 2019), LMs can significantly benefit downstream tasks. Furthermore, recent progress on generative LMs (Radford et al., 2019; Raffel et al., 2020; Lewis et al., 2020) suggests that it is possible to unify different tasks via one LM. The remarkable success

---
[*]Corresponding Authors.

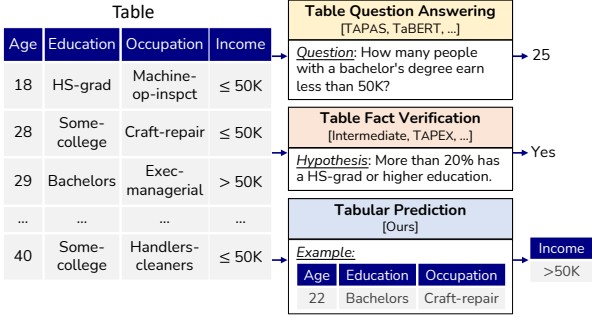

Figure 1: An illustration of different table-related tasks with representative table pre-training models, including TAPAS (Herzig et al., 2020), TaBERT (Yin et al., 2020), Intermediate (Eisenschlos et al., 2020), TAPEX (Liu et al., 2022) and our TAPTAP.

of pre-trained LMs has inspired much research in pre-training over structured tables, one of the most common types of data used in real-world applications (Benjelloun et al., 2020). Different from text, tables usually contain rich and meaningful structural information, and thus LMs on text corpus are not well suited for tabular data. To this end, there has been a growing amount of recent work on table pre-training (Herzig et al., 2020; Yin et al., 2020; Wang et al., 2021b; Liu et al., 2022).

However, the vast majority of existing table pre-training works aim to enhance joint reasoning over text and table (e.g., table question answering, tableQA), while neglecting tabular prediction, an important task in real-world applications. The goal of tabular prediction is to predict a specified target (e.g., the income) based on a set of features (e.g., the age and the occupation). As illustrated in Figure 1, most pre-trained LMs on tables such as TAPAS (Herzig et al., 2020) typically apply MLM variants on crawled tables and text segments to boost their joint reasoning capability in tableQA.

Nevertheless, as of yet, there is little evidence that these table pre-training methods can enhance the performance of tabular prediction tasks. This is probably because tabular prediction tasks are

quite challenging. In contrast to the exceptional performance of deep learning in many domains, recent studies (Shwartz-Ziv and Armon, 2022; Gorishniy et al., 2021) question the necessity of deep learning models for tabular prediction, as their performance is usually outperformed by traditional machine learning models. To summarize, it is still an open challenge to employ table pre-training to boost models for the tabular prediction task.

In this paper, we present TAPTAP (**Ta**ble **P**re-training for **Ta**bular **P**rediction), which is the first attempt that leverages pre-training of language models on tables to significantly benefit tabular prediction tasks. To benefit different backbone models, we apply table pre-training from a data perspective, i.e., we utilize TAPTAP to *synthesize high-quality examples* that can be used to train backbone models. Based on the widely used generative language model GPT (Radford et al., 2019), after ongoing pre-training on a large-scale corpus of real-world tabular data, TAPTAP is expected to capture a generic tabular data distribution. Then, TAPTAP can be quickly adapted to downstream tables via fine-tuning and can generate high-quality synthetic tables to support various applications on tabular data, including *privacy protection, low resource regime, missing value imputation*, and *imbalanced classification*. Meanwhile, such a design decouples the backbone model from the pre-trained model architecture, allowing TAPTAP to benefit different backbone models. Extensive experiments on 12 public datasets demonstrate that generative table pre-training can empower models on tabular prediction in various ways, and TAPTAP outperforms a total of 16 baselines in different scenarios and supports three state-of-the-art (SOTA) backbone models. The contributions of this paper can be summarized as follows:

- To our knowledge, we are the first to successfully apply table pre-training of language models to tabular prediction. With carefully designed generation strategies, our method combines the advantages of backbone models for tabular prediction and pre-trained LMs.

- To accomplish the pre-training, we collect and filter out 450 public tabular datasets from Kaggle, UCI, and OpenML platforms, and finally construct a large-scale pre-training corpus.

- To systematically evaluate the proposed table pre-training method, we build a comprehensive benchmark covering four practical settings in tabular prediction. Experimental results on the benchmark demonstrate that TAPTAP can be easily combined with different SOTA backbone models and outperforms a total of 16 baselines across 12 datasets.

## 2 Related Work

**Table Pre-training** Previous works on table pre-training can be categorized by the applicable downstream tasks and can be divided into four lines (Dong et al., 2022): *table question answering* which outputs the answer for questions over tables (Yin et al., 2020; Herzig et al., 2020; Yu et al., 2021; Liu et al., 2022; Andrejczuk et al., 2022), *table fact verification* which verifies whether a hypothesis holds based on the given table (Eisenschlos et al., 2020), *table to text* which generates textual descriptions from the given table (Gong et al., 2020; Xing and Wan, 2021) and *table structure understanding* which aims at identifying structural types in the given table (Tang et al., 2021; Wang et al., 2021b; Deng et al., 2021). Our work is different from theirs because we focus on the application of table pre-training on tabular prediction. There were some previous studies that performed tabular pre-training for tabular prediction (Wang and Sun, 2022; Arik and Pfister, 2021; Yoon et al., 2020; Bahri et al., 2022). The major difference between TAPTAP and the previous studies is that, TAPTAP performs cross-table pre-training on language models using a large number of tables to leverage the knowledge embedded in language models, and previous works usually perform single-table pre-training (Arik and Pfister, 2021) (or few tables with lots of overlapped columns (Wang and Sun, 2022)) on models specifically designed for tabular data.

**Table Generation** TAPTAP supports backbone models by generating synthetic tables, and thus it is close to the line of table generation. Existing methods for the generation of synthetic tabular data mostly leverage generative adversarial networks (Choi et al., 2017; Park et al., 2018; Mottini et al., 2018; Xu et al., 2019; Koivu et al., 2020) or variational autoencoders (Xu et al., 2019; Ma et al., 2020; Darabi and Elor, 2021). However, it is hard for these methods to leverage the textual semantics in tables. More recently, GReaT (Borisov et al., 2022) has successfully applied LMs in generating synthetic tabular data. There are some significant differences between GReaT and TAPTAP:

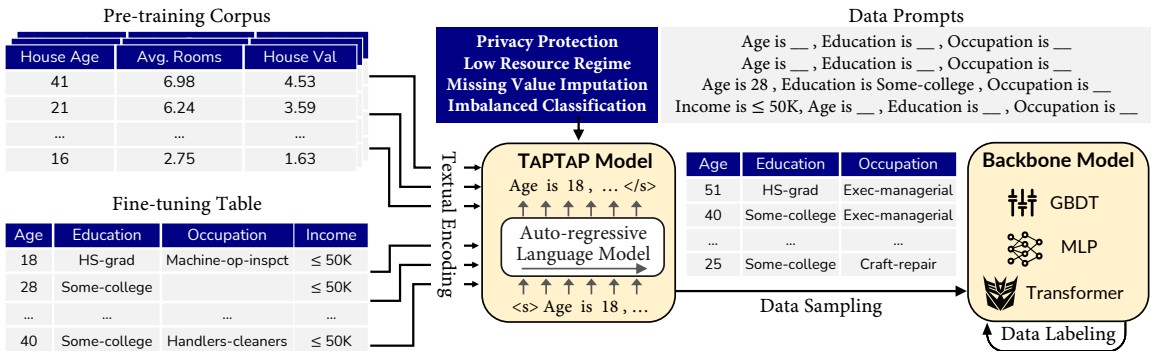

Figure 2: The illustration of our method. The TAPTAP model is firstly pre-trained on the pre-training corpus, and then fine-tuned on the downstream table. During both pre-training and fine-tuning, tables are serialized into sequences via textual encoding, and TAPTAP is trained to predict them token by token. During inference, TAPTAP is prompted to sample values for "__" in data prompts, and the filled values build up a synthetic table. Finally, once the backbone model has yielded labels for the synthetic table, it can be used to strengthen the backbone model.

1) We successfully apply table pre-training, which brings significant improvements over GReaT. 2) We design a delicate pre-training and data synthesis procedure, including number encoding and data labeling, which greatly benefit TAPTAP. 3) GReaT only exploits existing LMs for the privacy protection setting, along with basic backbone models like linear regression and decision trees. However, our proposed table pre-training can notably enhance the performance of three SOTA backbone models for tabular prediction across a range of scenarios.

**Tabular Prediction** Due to the tremendous success of deep learning and transfer learning in various domains, there has been a lot of research interest to extend this success to tabular prediction (Song et al., 2019; Wang et al., 2021a; Arik and Pfister, 2021). As for deep learning, we refer readers to Gorishniy et al. (2021) for a comprehensive comparison of different deep learning models for tabular data. Our work is technically orthogonal to these efforts, as it can be integrated with different backbone models (e.g., Transformer). As for transfer learning, there has been some pioneering attempts (Levin et al., 2022; Wang and Sun, 2022). More recently, researchers even explore the ability of LMs on zero / few-shot classification of tabular data (Hegselmann et al., 2022). However, there is often some gap between their experimental setup and real-world applications. For example, Levin et al. (2022) only investigates transfer learning on tables with lots of overlapping columns. Despite all these efforts in advancing deep learning on tabular data, recent studies (Shwartz-Ziv and Armon, 2022; Gorishniy et al., 2021) found that machine learning models like XGBoost (Chen and Guestrin, 2016)

and LightGBM (Ke et al., 2017) still outperformed those deep-learning counterparts. Therefore, TAP-TAP aims at synthesizing high-quality examples, which is able to empower both machine learning and deep learning models.

## 3 Methodology

### 3.1 Preliminary of Tabular Prediction

A tabular data usually contains two parts, the *features* and the *label*. Given the features as the input, the goal of tabular prediction is to predict the label. Taking the example from Figure 1, the task is to predict the income (label) of a person based on her / his age, education and occupation (features). Below we formalize tabular prediction using the binary-classification task, and the formulation can be easily extended to multi-class classification or regression problems. Formally, a tabular data with $n$ samples (i.e., rows) and $m$ features (i.e., columns) can be represented by $D = \{(\mathbf{x}_i, y_i)\}_{i=1,\ldots,n}$ where $\mathbf{x}_i = (x_{i,1}, \cdots, x_{i,j}, \cdots, x_{i,m}) \in \mathbb{R}^m$ and $y_i \in \{0, 1\}$. The $j$-th feature has a feature name $f_j$ (e.g., "age"). A model $F$ takes the features $\mathbf{x}_i$ as input to predict the label $y_i$. Our goal is to train a model such that the test error is as small as possible.

Existing works on improving $F$ either design better model architectures (Gorishniy et al., 2021) or improve the quality of training data (Zhang et al., 2022). We follow the second path to improve the model performance by generating synthetic data. There are four typical scenarios where high-quality synthetic samples are helpful: (1) **Privacy protection** (Gascón et al., 2017). In many application domains, each party only has part of the dataset and several parties can collaboratively train a model on

a joint dataset. But tabular data usually contains sensitive personal information or confidential business secrets that cannot be directly shared with other parties. In this case, TAPTAP can be used to generate synthetic data $D_s$ to replace the real data $D$, while achieving similar model performance. (2) **Low resource regime.** Data collection can be very expensive in some applications and hence handling the small data regime is an important challenge. For example, over $44\%$ classification datasets on the UCI platform (Asuncion and Newman, 2007) have less than 1000 samples. In this case, we can leverage TAPTAP to perform data augmentation in order to boost the backbone model. (3) **Missing value imputation.** Missing values are ubiquitous in tabular data (Stekhoven and Bühlmann, 2012). In this case, TAPTAP is able to impute the missing values to improve the performance of the model. (4) **Imbalanced classification.** It is common to have a long-tail label distribution in tabular data (Cao et al., 2019). In this case, TAPTAP can be used to balance the class distribution by conditional sampling (from the minority classes).

## 3.2 Overview

As shown in Figure 2, TAPTAP consists of four steps. (1) **Pre-training**: train an auto-regressive LM on the table pre-training corpus compiled by lots of public tabular datasets. (2) **Fine-tuning**: train the LM on the downstream table; (3) **Data Sampling**: prompt the fine-tuned LM to sample synthetic tables with only tabular features. (4) **Data Labeling**: assign pseudo labels to the synthetic tables via downstream backbone models. Below we describe these steps in details.

## 3.3 Pre-training

**Corpus Construction** To build the pre-training corpus, we leverage publicly available tabular datasets from Kaggle[1], UCI (Asuncion and Newman, 2007), and OpenML (Vanschoren et al., 2013) platforms. We believe the table pre-training should be performed on tabular data with rich semantic information, therefore we eliminate datasets with meaningless column names (e.g., V1). After the filtering, we finally collect $450$ tabular datasets with a total of nearly 2 million samples. To illustrate it better, we show in Figure 3 a word cloud composed of feature names and feature values. Note that we are careful to guarantee that the tabular

[1] https://www.kaggle.com/

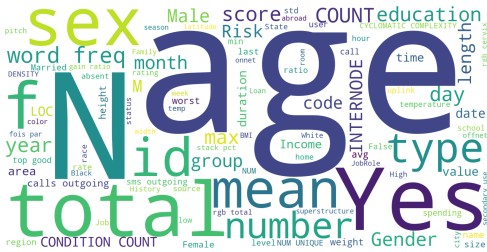

Figure 3: The word cloud for the pre-training corpus.

datasets used in pre-training and the downstream benchmark datasets are non-overlapping, so there is no data leakage issue.

### 3.3.1 Textual Encoding

**Table Serialization** Since TAPTAP starts with the GPT model, we follow the previous work (Borisov et al., 2022; Liu et al., 2022) to serialize each sample into a sequence of tokens to reduce the difficulty of table pre-training. As suggested by Hegselmann et al. (2022), we take the text template serialization strategy and serialize samples using the "[Feature] is [Value]" template. Taking the example in Figure 2, the first sample in the fine-tuning table is converted into a sentence "*Age is 18, Education is HS-grad, Occupation is Machine-op-inspct, Income is ≤ 50K*". Formally, given a table $D = \{(\mathbf{x}_i, y_i)\}$, let $x_{i,j}$ be the $j$-th feature value in $\mathbf{x}_i$ and $f_j$ be the $j$-th feature name. The textual encoding is to transform the $i$-th sample $\mathbf{x}_i$ into a splice of sentences separated by commas $\mathbf{t}_i = (t_{i,1}, ",", t_{i,2}, \cdots, ",", t_{i,m})$, where $t_{i,j} = (f_j, "is", x_{i,j})$.

**Number Encoding** Numerical features (e.g., age) are important and widely used in tabular data - over 70% of features in our pre-training corpus are numerical features, but how to properly encode these features has always been neglected in previous work of tabular prediction. Meanwhile, recent studies on LMs show that they are not good at dealing with numbers (Pi et al., 2022) and suggest the character-level representation is better suited to capture the number semantics than its counterparts (Wallace et al., 2019). Therefore, we use the character-level representation for all numerical features, which means that the phrase "*Age is 18*" in Figure 2 would be converted into "*Age is 1 8*".

**Permutation Function** The features in the tabular data are not ordered, but they are encoded as an ordered sentence, which introduces spurious positional relationships in textual encoding. In order to reconstruct the order independence among

Table 1: Properties of benchmark datasets.

| Dataset | Classification | | | | | | | Regression | | | | |
|---|---|---|---|---|---|---|---|---|---|---|---|---|
| | LO | AD | HE | CR | SI | BE | DI | CA | GE | ME | AG | DU |
| # samples (k) | 0.6 | 49 | 9.9 | 150 | 3.8 | 14 | 102 | 21 | 27 | 1.3 | 3.9 | 1.9 |
| # numerical features | 5 | 6 | 23 | 10 | 6 | 16 | 8 | 8 | 6 | 3 | 7 | 34 |
| # categorical features | 6 | 8 | 0 | 0 | 22 | 0 | 39 | 0 | 3 | 3 | 1 | 2 |
| # classes | 2 | 2 | 2 | 2 | 2 | 7 | 3 | - | - | - | - | - |

features, we follow previous work (Borisov et al., 2022) to apply a permutation function $\mathcal{P}$ to randomly shuffle the order of features when encoding a table. Therefore, the encoded sentence becomes $\mathbf{t}_i = (t_{i,k_1}, \text{","}, t_{i,k_2}, \cdots, \text{","}, t_{i,k_m})$, where $[k_1, k_2, \cdots, k_m] = \mathcal{P}([1, 2, \cdots, m])$. Such permutation enables *conditional sampling* when doing inference on downstream tables (Borisov et al., 2022), i.e., TAPTAP can generate a synthetic sample conditioned on any set of known features. We take a step further to demonstrate that the conditional sampling helps TAPTAP perform well in the missing value imputation scenario.

### 3.3.2 Pre-training Procedure

As mentioned before, the pre-training follows an auto-regressive manner, i.e., TAPTAP is trained to predict the encoded sentence token by token. Assuming we have $q$ tabular datasets for pre-training, the whole pre-training corpus $\mathcal{T}$ can be obtained by combining each tabular data after textual encoding as $\{\mathbf{t}_i^{(1)} \cup \cdots \cup \mathbf{t}_i^{(q)}\}$. Then, each sentence $\mathbf{t} \in \mathcal{T}$ can be encoded into a sequence of tokens using $(w_1, \cdots, w_N) = \texttt{tokenize}(\mathbf{t})$. In general, TAPTAP factorizes the probability of generating $\mathbf{t}$ in an auto-regressive manner as $p(\mathbf{t}) = \prod_{k=1}^{N} p(w_k | w_1, \cdots, w_{k-1})$. During pre-training, TAPTAP is optimized towards maximizing the probability $\prod_{i=1}^{|\mathcal{T}|} p(\mathbf{t}_i)$ on the entire pre-training corpus. The pre-training can start with any auto-regressive LM such as GPT (Radford et al., 2019), so that TAPTAP can benefit from the common knowledge already learned by these LMs.

### 3.4 Fine-tuning

Fine-tuning TAPTAP on the downstream table follows a similar procedure as in pre-training. The only difference is that the encoded sentences for fine-tuning are generated by applying textual encoding to the downstream table.

### 3.5 Data Sampling

Given the sequence $(w_1, \cdots, w_{k-1})$ as the prompt, TAPTAP is able to output the categorical distribution of the next token $w_k \in \mathcal{V}$ after fine-tuning, where $\mathcal{V}$ denotes the vocabulary. In general, $w_k$ is sampled from the conditioned probability distribution $p(w_k | w_1, \cdots, w_{k-1})$.

Since we also employ permutation during fine-tuning, the fine-tuned TAPTAP is able to generate synthetic samples given any prompt. We employ three kinds of prompting strategies for different application scenarios (Borisov et al., 2022). **(1) Feature name as prompt**. This strategy is used in the privacy protection and low resource regime, where only feature names in the tabular data are selected as the prompt. The synthetic samples are generated by TAPTAP according to the prompt "[Feature] is ". **(2) One feature-value pair as prompt**. This strategy is used in the imbalanced classification scenario, where the feature names and the minority label(s) are both provided as the prompt. With the label treated as a feature, TAPTAP generates synthetic samples based on the prompt "[Feature] is [Value], ". **(3) Multiple feature-value pairs as prompt**. This strategy is used in the missing feature scenarios, where the feature names and available feature values are provided as the prompt. TAPTAP generates synthetic samples according to the prompt "[Feature1] is [Value1], [Feature2] is [Value2], $\cdots$, ". The order of the given features in the prompt is random. Data prompt examples can be found in Figure 2.

### 3.6 Data Labeling

An accurate label is arguably one of the most crucial ingredients in synthetic samples. Noisy labels can severely degrade the generalization capability of backbone models (Gorishniy et al., 2021). In contrast to the previous work relying on LMs to generate labels (Borisov et al., 2022), we propose to assign pseudo labels using the SOTA backbone models. We argue that LMs are not yet the best

Table 2: The experimental results in **privacy protection**. Here we present the difference in metrics between the model trained on the synthetic data and the one trained on the original data, the lower the better. A gap close to zero suggests that the synthetic data is of comparable quality to the original data. Below the backbone model is LightGBM. Results of MLP and Transformer can be found in Table 13 and 14.

| Diff. *w.r.t.* Origin ↓ | LO | AD | HE | CR | SI | BE | DI | CA | GE | ME | AG | DU | Avg. Rank |
|---|---|---|---|---|---|---|---|---|---|---|---|---|---|
| CTGAN | 12.1 | 3.7 | 8.6 | 87.2 | 4 | 24.2 | 5.6 | 38.4 | 14.3 | 101.1 | 27.5 | 104.1 | 5.88 ± 0.91 |
| CopulaGAN | 14.2 | 3.4 | 8.7 | 0.2 | 4 | 27.8 | 5.5 | 57.3 | 14.5 | 83.6 | 26.6 | 105.7 | 5.79 ± 1.47 |
| TVAE | 14 | 5.7 | 0.8 | 1.4 | 7.6 | 7.9 | 2.6 | 16.7 | 5.1 | 109.4 | 33.7 | 34.2 | 5.00 ± 1.41 |
| GReaT-distill | 1.4 | 2 | 2.8 | 2.4 | 19.8 | 1.8 | 4.8 | 22.8 | 8.1 | 8.4 | 7.6 | 87.7 | 4.50 ± 1.09 |
| GReaT | 2.5 | 0.9 | 3.7 | 2.6 | 14.5 | 1.9 | 1.7 | 13.1 | 2.4 | 0.7 | 4.5 | 25.7 | 3.75 ± 1.29 |
| TAPTAP-distill | 0 | 0.7 | 0.4 | 0 | 1.1 | 0.4 | 1.6 | 3.7 | 0.7 | 0.2 | 0.6 | 16.7 | 1.71 ± 0.45 |
| TAPTAP | 0 | 0.5 | 0.3 | 0 | 0.6 | 0.4 | 1.6 | 2.5 | 1.5 | 0 | 4.6 | 12.8 | 1.38 ± 0.64 |

Table 3: The experimental results in **low resource regime**. "+ Ori" means training with the original data. "+ Ori + Synthetic Data" means training with the original data plus the synthetic data. Below the backbone model is Transformer with piece-wise linear encoding. The full results on all datasets can be found in Table 15 and 16.

| Metric ↑ | LO | HE | BE | SI | CA | GE | ME | AG | DU | Avg. Rank |
|---|---|---|---|---|---|---|---|---|---|---|
| Transformer + Ori | 76.8 | 72.5 | 92.7 | 98.5 | 82.9 | 98.2 | 86.6 | 52.6 | 96.5 | - |
| *Transformer + Ori + Synthetic Data by Models* | | | | | | | | | | |
| CTGAN | 74.7 | 71.5 | 92.7 | 97.8 | 81.5 | 96.3 | 72.1 | 51.6 | 71.7 | 5.83 ± 1.27 |
| CopulaGAN | 74.7 | 71.8 | 92.5 | 97.8 | 81.7 | 95.9 | 72.8 | 52.0 | 86.8 | 5.39 ± 1.32 |
| TVAE | 76.2 | **72.8** | 92.5 | 97.4 | 82.0 | 97.2 | 85.7 | 47.3 | 80.0 | 4.56 ± 2.01 |
| GReaT-distill | 76.1 | 72.0 | 92.6 | 98.3 | 77.9 | 96.6 | 86.2 | 52.4 | 79.0 | 4.89 ± 1.05 |
| GReaT | 74.5 | 72.1 | 92.7 | 98.4 | 80.5 | 98.1 | 86.4 | 53.3 | 80.3 | 4.11 ± 1.45 |
| TAPTAP-distill | 76.2 | 72.5 | 92.8 | **98.5** | **83.7** | **98.2** | **86.9** | **53.8** | **98.2** | 1.78 ± 0.67 |
| TAPTAP | **77.5** | 72.5 | **92.9** | **98.5** | **83.7** | **98.2** | 86.7 | 53.5 | 97.9 | 1.44 ± 0.53 |

choice for label generation, since most commonly used tabular prediction models (e.g., LightGBM) are carefully designed for tabular data and generally more accurate at predicting the labels (Hegselmann et al., 2022; Shwartz-Ziv and Armon, 2022).

Formally, given a downstream table $D = \{(\mathbf{x}_i, y_i)\}$, we first fine-tune TAPTAP on it to generate synthetic tabular features $\{\mathbf{x}_i'\}$. Next, a backbone model $F$ is trained to fit the original table $D$. Then, the synthetic labels $y_i'$ can be derived using the well-trained model via $y_i' = F(\mathbf{x}_i')$. Finally, the synthetic labels and the synthetic tabular features make up the final synthetic table $D_s = \{(\mathbf{x}_i', y_i')\}$. The following model analysis in the Section 4.3 reveals that our design of data labeling (i.e., not using LMs for label generation) is crucial for the superior performance of our approach.

## 4 Experiments

### 4.1 Experimental Setup

**Datasets and Evaluation Metrics** We collect 12 diverse real-world datasets from various domains (Asuncion and Newman, 2007; Vanschoren et al., 2013). Each dataset is split into a train set (75%) and a test set (25%), and all experiments

share the same splits. We provide some important statistics of each dataset in Table 1 and more details in Appendix A. Following previous works (Grinsztajn et al., 2022; Borisov et al., 2022), we use accuracy and R2 score as the evaluation metrics for the classification and regression tasks. For the imbalanced classification scenario, we employ AUC as the evaluation metric. All the experimental results are averaged over 10 different random seeds.

**Backbone Models** To comprehensively evaluate TAPTAP, we experiment with various SOTA backbone models for tabular prediction, including LightGBM (Ke et al., 2017), MLP, and Transformer (Gorishniy et al., 2021). Modern GBDT models (such as LightGBM and XGBoost) have been the most popular models for tabular prediction in the past few years (Shwartz-Ziv and Armon, 2022). We choose LightGBM in our experiments. Recently, MLP and Transformer with piece-wise linear encoding (Gorishniy et al., 2022) are proposed to be competitive against LightGBM.

**Language Models.** TAPTAP uses the original GPT2 (Radford et al., 2019) with 355M parameters, while TAPTAP-distill uses the distilled version of

GPT2 (Sanh et al., 2019) with 82M parameters.

## 4.2 Main Results

We measure the quality of the synthesized samples by their performance in the application scenarios.

**Privacy Protection**  Following the previous work (Borisov et al., 2022), we include baselines CT-GAN (Xu et al., 2019), TVAE (Xu et al., 2019), CopulaGAN (Patki et al., 2016), GReaT-distill and GReaT (Borisov et al., 2022). All methods are used to generate the same amount of synthetic data as the original dataset. The backbone models are trained on the synthetic data, and then evaluated on the original test set. The experimental results are presented in Table 2. One can observe that TAPTAP and TAPTAP-distill outperform most of the baseline methods. Noticing that GReaT also utilizes GPT2, the fact that TAPTAP surpasses it by a large margin suggests the superiority of table pre-training. More importantly, with table pre-training, the quality of the synthetic data generated by TAP-TAP can even match that of the original data. On half of the privacy protection datasets, LightGBM models trained with our synthetic data achieve almost the same performance as with the original data. This is highly impressive, especially when considering that none of the synthetic samples appear in the original dataset.

**Low Resource Regime**  We perform data augmentation to mitigate the low resource dilemma. The baseline methods are identical to those in privacy protection. During fine-tuning, following the experience of multi-task learning in T5 (Raffel et al., 2020), we first use the synthetic data to fine-tune a backbone model. Then, we use the original data to continually fine-tune the model. Experimental results on 9 datasets with less than 30k samples are presented in Table 3, which show that TAPTAP is able to perform comparably or better than all baseline methods on most datasets. Furthermore, TAPTAP contribute significant gains to 4 of the 9 datasets, which is highly non-trivial.

**Missing Value Imputation**  We compare with top methods as baselines in a recent benchmarking study (Jarrett et al., 2022), including GAIN (Yoon et al., 2018), HyperImpute (Jarrett et al., 2022), MICE (Van Buuren and Groothuis-Oudshoorn, 2011), MissForest (Stekhoven and Bühlmann, 2012), MIWAE (Mattei and Frellsen, 2019), and Sinkhorn (Muzellec et al., 2020). Following previ-

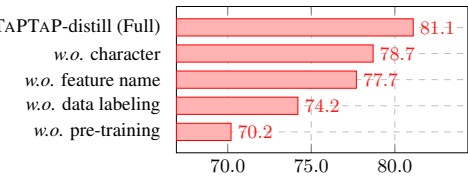

Figure 4: Experimental results in the ablation study. The y-axis is the average metric values across all datasets in the privacy protection setting with LightGBM.

ous work (Jarrett et al., 2022), two missing mechanisms are used to yield missing values: missing completely at random (MCAR) and missing at random (MAR). The miss ratio is set to be 0.3. We present the results in Table 4. As observed, TAP-TAP always outperforms most baseline methods using one LM and receives the highest average ranking, indicating its superiority.

**Imbalanced Classification**  By generating synthetic samples for the minority class, TAPTAP addresses the imbalanced classification problem. Therefore, we compare our methods against popular oversampling methods (Camino et al., 2020), including Random, SMOTE (Chawla et al., 2002), ADASYN (He et al., 2008), Borderline (Han et al., 2005), SMOTE+ENN (Alejo et al., 2010) and SMOTE+Tomek (Zeng et al., 2016). Following the standard approach (Buda et al., 2018), we down-sample the minority class of each binary-classification dataset so that the imbalanced ratio is 50 (#majority / #minority = 50). Experimental results on five binary-classification datasets are presented in Table 5, where TAPTAP still has the highest average ranking among all baselines.

**Overall Summarization**  First, TAPTAP generally improves the performance of different backbone models in tabular prediction and outperforms the majority of baseline methods on various tabular prediction scenarios. Second, the advantage of TAPTAP over TAPTAP-distill suggests that table pre-training can also benefit from scaling up LMs. Third, TAPTAP is the first to successfully generate synthetic data for comparable backbone model performance to original data.

## 4.3 Ablation Study

To investigate the effectiveness of each component in TAPTAP, we conduct an ablation study. We name TAPTAP without different components as follows: (1) **w.o. pre-training** refers to TAPTAP without table pre-training. (2) **w.o. data labeling**

Table 4: The experimental results in **missing value imputation**. "+ M-Ori" means training with the original data processed by the MCAR mechanism. "+ M-Ori + Synthetic Data" means training with the M-Ori data where the missing values are imputed by different models. Below the backbone model is MLP. Results using LightGBM and Transformer as backbone models can be found in Table 17 and 18. Results with the MAR mechanism can be found in Appendix B.6. ✗ denotes the method cannot run successfully on the dataset due to too many missing values.

| Metric ↑ | LO | AD | HE | CR | SI | BE | DI | CA | GE | ME | AG | DU | Avg. Rank |
|---|---|---|---|---|---|---|---|---|---|---|---|---|---|
| MLP + M-Ori | 73.2 | 85.4 | 71.1 | 93.6 | 95.6 | 90.3 | 57.4 | 63.1 | 93.0 | 68.4 | 41.4 | 70.6 | - |
| *MLP + M-Ori + Synthetic Data by Models* | | | | | | | | | | | | | |
| MIWAE | 71.3 | ✗ | 68.7 | ✗ | 95.7 | 90.0 | ✗ | 59.0 | 90.6 | 65.0 | 42.6 | 75.7 | 7.54 ± 0.78 |
| Sinkhorn | 73.2 | 83.9 | 69.3 | 93.5 | 95.8 | 88.6 | 56.6 | 62.5 | 93.4 | 73.3 | 49.9 | 72.8 | 5.75 ± 1.14 |
| GAIN | **86.2** | 86.2 | 72.9 | 57.6 | **97.6** | 90.9 | 53.8 | 54.9 | 92.9 | 69.4 | 44.2 | 82.0 | 5.00 ± 2.49 |
| MICE | 73.1 | 84.5 | 70.0 | 93.6 | 96.0 | 88.3 | 57.2 | 63.0 | 93.8 | 72.3 | 53.1 | **91.1** | 4.75 ± 1.82 |
| MissForest | 72.9 | 79.8 | 69.9 | 92.7 | 96.7 | 91.6 | 57.2 | 74.0 | 94.2 | 79.5 | 46.4 | 88.5 | 4.75 ± 1.54 |
| HyperImpute | 73.4 | 86.7 | 70.5 | 83.0 | 96.8 | 92.8 | ✗ | 77.7 | 96.2 | 78.4 | **58.1** | 90.6 | 3.54 ± 1.78 |
| TAPTAP-distill | 74.9 | 86.9 | 72.2 | **93.7** | 97.5 | **93.4** | 57.2 | 78.5 | 94.5 | 72.6 | 53.6 | 69.2 | 2.83 ± 1.90 |
| TAPTAP | 73.6 | **87.0** | **73.0** | **93.7** | 96.9 | 93.1 | **57.8** | **82.7** | **97.3** | **81.2** | 53.2 | 85.5 | 1.83 ± 1.11 |

Table 5: Experimental results in **imbalanced classification**. "I-Ori" is the imbalanced data. Below the backbone model is LightGBM. ✗ denotes the method cannot run successfully on the dataset due to too few samples in the minority class. The metric is AUC.

| Metric ↑ | LO | AD | HE | CR | SI | Avg. Rank |
|---|---|---|---|---|---|---|
| LightGBM + I-Ori | 71.2 | 90.2 | 82.3 | 84.0 | 99.4 | - |
| *LightGBM + I-Ori + Synthetic Data by Models* | | | | | | |
| SMOTE+ENN | ✗ | 87.9 | 77.7 | 83.7 | 98.9 | 7.30 ± 0.97 |
| SMOTE+Tomek | ✗ | 89.3 | 80.3 | 84.1 | 99.5 | 5.80 ± 1.44 |
| ADASYN | ✗ | 89.5 | 79.6 | 84.0 | 99.5 | 5.30 ± 0.97 |
| Random | 51.7 | 89.4 | 82.3 | 82.9 | 99.7 | 5.00 ± 2.00 |
| SMOTE | ✗ | 89.5 | 80.3 | 84.1 | 99.5 | 5.00 ± 1.37 |
| Borderline | 71.2 | 89.6 | 79.3 | 83.5 | **99.8** | 4.20 ± 2.68 |
| TAPTAP-distill | 73.0 | **91.3** | **83.8** | 84.8 | 99.7 | 1.80 ± 0.45 |
| TAPTAP | **85.5** | **91.3** | 83.0 | **85.0** | 99.7 | 1.60 ± 0.89 |

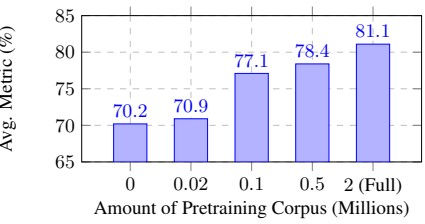

Figure 5: The influence of pre-training scale on the downstream performance. The value of each method is the average metric values across all datasets in the privacy protection setting with LightGBM.

Table 6: The comparison between TAPTAP and TAP-TAP with additional web tables for pre-training.

|  | AD | HE | CR | SI | DI | CA |
|---|---|---|---|---|---|---|
| TAPTAP | 87.5 | 72.2 | 93.8 | 97.6 | 57.8 | 81.5 |
| + Web Tables | 87.7 | 72.3 | 93.8 | 98.2 | 57.6 | 82.1 |

refers to TAPTAP using LMs to generate labels. (3) *w.o.* **character** refers to TAPTAP without using the character-level representations for numerical features. (4) *w.o.* **feature name**. The column names of each dataset are replaced by dummy names (e.g., "V1") to remove semantic information.

The experimental results are visualized in Figure 4. We present the average metric values (i.e., Acc. or R2) of each method across 12 datasets in the privacy protection setting, since it is the most straightforward setting to indicate the quality of synthetic data. We can see that pre-training and data labeling are particularly important for TAPTAP. The semantic information in column names and the character-level representation to enhance number encoding also provide considerable improvement.

## 4.4 Analysis

**The Scale of Pre-training Corpus** Figure 5 illustrates the influence of the pre-training scale on the downstream performance. We present the results with 0.02, 0.1, 0.5 and 2 million samples. As one can observe, scaling up the pre-training corpus brings positive effects. However, the number of high-quality real-world tabular datasets is limited. Therefore, it may be helpful to take advantage of the millions of tables available on the Web.

**Pre-training using Web Tables** To explore the above direction, we present a preliminary study on using tables from Web for pre-training. We parse over 130k Web tables with a total of 8 million samples from the WikiTables corpus (Bhagavatula et al., 2015). We use the Web tables together with

the tabular datasets for pre-training. The results of the privacy protection setting are presented in Table 6. We can see that even with a large number of Web tables, it is still hard to further boost the backbone models. We attribute it to the quality issue. The collected tabular datasets have already been examined by the platforms, and usually have higher quality than noisy Web tables. How to automatically identify high-quality tables from the huge number of Web tables for pre-training is a promising future direction.

## 5 Conclusion & Future Work

In this paper, we propose TAPTAP, a table pre-training method to empower models for tabular prediction. It can be combined with various backbone models and boost them via synthesizing high-quality tabular data. A large-scale empirical study demonstrates that TAPTAP can benefit different SOTA backbone models on four tabular prediction scenarios. In the future, we plan to extend TAPTAP to process tables with a large number of features.

## 6 Acknowledgement

We thank all the anonymous reviewers for their constructive feedback and insightful comments. Tianping Zhang, Shaowen Wang, and Jian Li are supported in part by the National Natural Science Foundation of China Grant 62161146004.

### Limitations

The major limitation of TAPTAP is the scalability. While we enjoy the advantages of LMs, we also introduce the drawbacks of LMs. In practice, TAPTAP usually requires more running time and GPU memory than other methods. Detailed comparison can be found in Appendix B.3. In addition, TAPTAP can only process tabular data with less than 100 features due to the input length limitation that GPT can process (i.e., 1024 tokens).

### Ethics Statement

In this paper, we collected and filtered out 450 publicly available tabular datasets to construct the pre-training corpus for TAPTAP. As these datasets have been reviewed by well-known machine learning platforms such as Kaggle, they should have no private information about individuals. However, we cannot confirm whether these datasets contain potential biases since the corpus contains

millions of samples. For example, there may be tables that have the potential to wrongly associate recruitment result to gender. Also, since our model is pre-trained based on GPT, readers may be concerned that the synthetic tables generated by our model contain offensive content. On this point, we argue that one might not be worried too much since for categorical features, our model can be easily tuned to only generate the values that appear in the downstream table, which is relatively controllable.

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

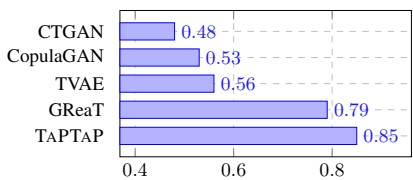

Figure 6: Sampling diversity in terms of the coverage score averaged across datasets.

# A  Datasets

We provide the urls of the public datasets in Table 7. These datasets are publicly available, and their license permits usage for research purposes.

# B  Additional Experiments

## B.1  Distance to Closest Record

In order to demonstrate that TAPTAP generates synthetic samples similar to the original data instead of copying the original data, following the standard approach (Borisov et al., 2022), we calculate each sample's distance to the closest record (DCR) in the original training data $D$. For each synthetic sample $x$, its DCR is $DCR(x) = min\{Distance(x, x_i)|x_i \in D\}$. We use the $L_1$ distance for numerical features. For categorical features, we set the distance to be 0 for equal categories and 1 otherwise. We present the results of California Housing and HELOC in Figure 7 and 8.

## B.2  Sampling Diversity

We employ the coverage score (Naeem et al., 2020) to quantitatively evaluate the sampling diversity of TAPTAP and baseline methods. The coverage refers to the proportion of actual records that contain at least one synthetic record within its manifold. A manifold is defined as a sphere surrounding the sample, with a radius of r determined by the distance between the sample and its k-th nearest neighbor. We present the averaged coverage score in Figure 6.

## B.3  Running Time Comparison

We analyze the running time of TAPTAP, TAPTAP-distill, and baseline methods. The experiments are carried out on a single NVIDIA GeForce RTX 3090 with 24 GB RAM, 64 system RAM, and Intel(R) Xeon(R) Platinum 8350C CPU @ 2.60GHz with 16 cores. For the privacy protection setting, we present the running time of training/fine-tuning and sampling separately. We present the results of the Adult Income dataset in Table 8. For the missing

Table 7: The urls of test datasets.

| Dataset | Link |
| --- | --- |
| Adult Income (AD) (Kohavi, 1996) | https://archive.ics.uci.edu/ml/datasets/Adult |
| HELOC (HE) | https://www.kaggle.com/datasets/averkiyoliabev/home-equity-line-of-creditheloc |
| California Housing (CA) (Pace and Barry, 1997) | https://www.kaggle.com/datasets/camnugent/california-housing-prices |
| Diabetes (DI) (Strack et al., 2014) | https://www.kaggle.com/c/1056lab-diabetes-readmission-prediction |
| Credit Scoring (CR) (Credit Fusion, 2011) | https://www.kaggle.com/competitions/GiveMeSomeCredit/overview |
| Loan (LO) | https://www.openml.org/search?type=data&status=active&sort=match&id=43595 |
| Dubai Housing (DU) | https://www.kaggle.com/datasets/dataregress/dubai-properties-dataset |
| Crab Age (AG) (Sidhu, 2021) | https://www.kaggle.com/datasets/sidhus/crab-age-prediction |
| Medical Cost (ME) | https://www.kaggle.com/datasets/mirichoi0218/insurance |
| Gem Price (GE) | https://www.kaggle.com/datasets/colearninglounge/gemstone-price-prediction |
| Bean Type (BE) (Koklu and Özkan, 2020) | https://archive.ics.uci.edu/ml/datasets/Dry+Bean+Dataset |
| Sick Record (SI) (Quinlan, 1987) | https://www.openml.org/search?type=data&sort=runs&id=38&status=active |

value imputation setting, we present the running time of the California Housing dataset in Table 9. We can see that TAPTAP and TAPTAP-distill requires more running time than most of the baseline methods. While we enjoy the benefits of leveraging LMs to achieve top performance, we also introduce the drawbacks of LMs in requiring more computational resources. However, there are important real-world applications such as healthcare or finance where achieving better performance outweighs saving computational time. In addition, the fine-tuning and sampling time can be reduced by using more computational resources.

### B.4 Privacy Protection

Table 2, 13 and 14 show the performance of our method and baseline methods in privacy protection setting with LightGBM, MLP, and Transformer as the backbone.

### B.5 Low Resource Regime

Table 15 and 16 show the performance of our method and baseline methods in low resource regime setting with MLP and Transformer as the backbone. Note that both low resource datasets and high resource datasets are presented in the table.

### B.6 Missing Value Imputation

Table 17, 4 and 18 show the performance of our method and baseline methods in missing value imputation setting using MCAR mechanism with LightGBM, MLP, and Transformer as the backbone. Table 19, 20 and 21 show the performance of our method and baseline methods in missing value imputation setting using MAR mechanism with LightGBM, MLP, and Transformer as the backbone. MIWAE and HyperImpute fail on some datasets because one feature in the dataset contains too many missing values. For example, 96.9% of

data points in the "weight" column in the Diabetes dataset are missing. However, the methods require at least one valid value for each training batch.

### B.7 Imbalance Classification

Table 5 shows the performance of our method and baseline methods in the imbalance classification setting with LightGBM as the backbone. Smote-based methods fail on the Loan dataset because there are fewer than 10 minority class data, which results in the number of sampled data points being less than the number of neighbors(Chawla et al., 2002) required.

## C Hyperparameters Optimization

We use optuna (Akiba et al., 2019) to tune the hyperparameters of our backbone models, i.e. LightGBM, MLP, and Transformer. For each specific dataset and model, we first use the original data to tune the hyperparameters of the model. Then the set of hyperparameters are used throughout all the experiments of the dataset on all the methods for a fair comparison.

### C.1 LightGBM

When tuning the hyperparameters of LightGBM, the following hyperparameters are fixed:

1. boosting = "gbdt"

2. early_stopping_round = 50

3. n_estimators = 1000

Other hyperparameters and the search space for tuning are in Table 10.

### C.2 MLP

We follow the implementation in Gorishniy et al. (2022). We present the hyperparameters space for searching in Table 11.

Table 8: The running time in seconds on the Adult Income dataset of different methods in the privacy protection setting. The number of fine-tuning steps for GReaT and TAPTAP was 10k. A total of 36k samples were generated.

| | CTGAN | CopulaGAN | TVAE | GReaT-distill | GReaT | TAPTAP-distill | TAPTAP |
|---|---|---|---|---|---|---|---|
| Training/Fine-tuning Time | 873 | 846 | 360 | 960 | 3770 | 910 | 3680 |
| Sampling Time | 9 | 11 | 3 | 895 | 1395 | 506 | 1185 |

Table 9: The running time in seconds on the California Housing dataset of different methods in the missing value imputation setting. The number of fine-tuning steps for GReaT and TAPTAP was 10k. A total of 15k samples were imputed.

| | MIWAE | HyperImpute | GAIN | MICE | MissForest | Sinkhorn | TAPTAP-distill | TAPTAP |
|---|---|---|---|---|---|---|---|---|
| Running Time | 210 | 175 | 8 | 336 | 47 | 565 | 1215 | 4008 |

Table 10: LightGBM hyperparameter space

| Parameter | Distribution |
|---|---|
| learning_rate | Uniform[0.01,0.05] |
| num_leaves | UniformInt[10,100] |
| min_child_weight | LogUniform[1e-5,1e-1] |
| min_child_samples | UniformInt[2,100] |
| subsample | Uniform[0.5,1.0] |
| colsample_bytree | Uniform[0.5,1.0] |
| # Iterations | 100 |

Table 11: MLP hyperparameter space

| Parameter | Distribution |
|---|---|
| # Layers | UniformInt[1,16] |
| Layer size | UniformInt[1,1024] |
| Dropout | Uniform[0,0.5] |
| Learning rate | {0, Uniform[0,0.5]} |
| Weight decay | LogUniform[5e-5,0.005] |
| # Iterations | 100 |

Table 12: Transformer hyperparameter space

| Parameter | Distribution |
|---|---|
| # Layers | UniformInt[1,4] |
| Embedding size | UniformInt[96,512] |
| Residual dropout | {0, Uniform[0,0.2]} |
| Attetion dropout | Uniform[0,0.5] |
| FFN dropout | Uniform[0,0.5] |
| FFN factor | Uniform[2/3,8/3] |
| Leaning rate | LogUniform[1e-5, 1e-3] |
| Weight decay | LogUniform[1e-6, 1e-4] |
| # Iterations | 100 |

## C.3 Transformer

We follow the implementation in Gorishniy et al. (2022). We present the hyperparameters space for searching in Table 12.

## D  Reproducibility Details

For the baseline methods of CT-GAN, TVAE, and CopulaGAN in the privacy protection and low resource regime setting, we use the implementation in `https://sdv.dev/SDV/user_guides/single_table/models.html`. For GReaT-distill and GReaT, we use the implementation in `https://github.com/kathrinse/be_great`. For the baseline methods of GAIN, HyperImpute, MICE, MissForest, MIWAE, Sinkhorn in the missing value imputation setting, we use the implementation in `https://github.com/vanderschaarlab/hyperimpute`. For the base-

line methods of Random, SMOTE, ADASYN, Borderline, SMOTE+ENN, SMOTE+Tomek in the imbalanced classification setting, we use the implementation in `https://github.com/scikit-learn-contrib/imbalanced-learn`.

We use the implementation of GPT2 and the distilled version of GPT2 in the huggingface platform (Wolf et al., 2019). We pre-train TAPTAP and TAPTAP-distill for 80,000 steps. We finetune the TAPTAP, TAPTAP-distill, GReaT, and GReaT-distill model for 10,000 steps, except for the Credit Scoring (CR) and Sick Records (SI) datasets, which we finetune the model for 20000 steps. The batch size is 64 for all the datasets. In the privacy protection, low resource regime, and imbalanced classification setting, we use the one feature-value pair as prompt sampling method. We start sampling with the target feature following the previous approach (Borisov et al., 2022). The missing value imputation setting does not require pseudo label generation, as the missing mechanism only drops the feature values and the labels are always provided. In the imbalanced classification setting, we generate synthetic samples on the minority class until the number of samples is the same for the minority and majority class.

Table 13: The experimental results in **privacy protection**. "+ Ori" means training with the original data. Below the backbone model is MLP with piece-wise linear encoding.

| Metric ↑ | LO | AD | HE | CR | SI | BE | DI | CA | GE | ME | AG | DU | Avg. Rank |
|---|---|---|---|---|---|---|---|---|---|---|---|---|---|
| MLP + Ori | 76.6 | 87.7 | 72.4 | 93.8 | 98.3 | 92.7 | 58.7 | 81.8 | 98.1 | 85.7 | 52.9 | 99.5 | - |
| *MLP + Synthetic Data by Models* | | | | | | | | | | | | | |
| CTGAN | 64.9 | 84.3 | 65.4 | 6.6 | 94.7 | 65.7 | 53.8 | 46.4 | 83.5 | -16.7 | 31.0 | -10.6 | 6.17 ± 0.62 |
| CopulaGAN | 63.2 | 84.1 | 64.0 | 93.5 | 94.7 | 62.3 | 53.9 | 31.2 | 83.6 | 10.3 | 31.4 | -15.4 | 5.96 ± 1.25 |
| TVAE | 64.9 | 82.7 | 71.7 | 92.4 | 95.7 | 70.9 | 56.1 | 60.8 | 93.8 | -22.0 | 19.0 | 66.2 | 5.04 ± 1.36 |
| GReaT-distill | 74.4 | 85.8 | 69.8 | 91.3 | 97.1 | 90.9 | 53.9 | 65.4 | 90.6 | 80.6 | 47.1 | 15.7 | 4.42 ± 1.00 |
| GReaT | 73.9 | 86.7 | 71.0 | 91.5 | 97.4 | 90.7 | **57.6** | 72.0 | 96.6 | 84.7 | 49.2 | 69.5 | 3.25 ± 0.97 |
| TAPTAP-distill | 77.0 | **87.4** | **72.3** | **93.8** | 97.5 | 92.2 | 57.2 | 80.5 | **98.1** | 86.7 | **53.3** | 77.6 | 1.83 ± 0.58 |
| TAPTAP | **77.1** | **87.4** | **72.3** | **93.8** | 97.8 | **92.6** | 57.3 | **81.9** | 97.1 | 85.2 | 51.6 | **86.4** | 1.33 ± 0.49 |

Table 14: The experimental results in **privacy protection**. "+ Ori" means training with the original data. Below the backbone model is Transformer with piece-wise linear encoding.

| Metric ↑ | LO | AD | HE | CR | SI | BE | DI | CA | GE | ME | AG | DU | Avg. Rank |
|---|---|---|---|---|---|---|---|---|---|---|---|---|---|
| Transformer + Ori | 76.8 | 87.4 | 72.5 | 93.8 | 98.5 | 92.7 | 58.7 | 82.9 | 98.2 | 86.6 | 52.6 | 96.5 | - |
| *Transformer + Synthetic Data by Models* | | | | | | | | | | | | | |
| CTGAN | 65.1 | 84.1 | 65.2 | 6.6 | 94.7 | 55.9 | 53.8 | 48.3 | 84.3 | -14.3 | 24.5 | -12.4 | 6.17 ± 0.58 |
| CopulaGAN | 61.3 | 84.2 | 64.3 | 93.5 | 94.6 | 58.0 | 53.9 | 30.5 | 83.3 | 11.1 | 23.2 | -16.1 | 6.08 ± 1.24 |
| TVAE | 65.6 | 82.5 | 71.7 | 92.0 | 95.6 | 65.7 | 56.2 | 58.9 | 92.8 | -24.2 | 19.1 | 49.8 | 5.00 ± 1.35 |
| GReaT-distill | 74.2 | 85.4 | 69.1 | 91.3 | 96.8 | 90.5 | 54.0 | 64.6 | 90.7 | 83.4 | 44.6 | 14.2 | 4.25 ± 0.87 |
| GReaT | 72.3 | 86.5 | 68.7 | 91.3 | 97.2 | 90.3 | **57.6** | 71.9 | 96.4 | 85.6 | 48.1 | 60.4 | 3.25 ± 1.14 |
| TAPTAP-distill | 76.7 | 87.2 | **72.2** | **93.8** | 97.3 | 92.0 | 57.2 | 81.5 | **98.1** | 86.9 | **53.5** | 63.1 | 1.75 ± 0.62 |
| TAPTAP | **77.1** | **87.3** | 72.1 | **93.8** | **98.1** | **92.5** | 57.3 | **83.1** | 96.0 | 86.1 | 51.4 | **75.2** | 1.50 ± 0.67 |

Table 15: The experimental results in **low resource regime**. "+ Ori" means training with the original data. "+ Ori + Synthetic Data" means training with the original data plus the synthetic data. Below the backbone model is MLP with piece-wise linear encoding.

| Metric ↑ | LO | AD | HE | CR | SI | BE | DI | CA | GE | ME | AG | DU | Avg. Rank |
|---|---|---|---|---|---|---|---|---|---|---|---|---|---|
| MLP + Ori | 76.6 | 87.7 | 72.4 | 93.8 | 98.3 | 92.7 | 58.7 | 81.8 | 98.1 | 85.7 | 52.9 | 99.5 | - |
| *MLP + Ori + Synthetic Data by Models* | | | | | | | | | | | | | |
| CTGAN | 76.4 | 87.4 | 71.1 | 84.8 | 96.2 | 93.0 | 58.5 | 80.2 | 95.6 | 69.2 | 50.3 | 83.9 | 6.08 ± 1.00 |
| CopulaGAN | 76.5 | 87.5 | 71.4 | **93.8** | 97.4 | 93.0 | 58.5 | 80.0 | 95.1 | 70.6 | 51.9 | 97.9 | 4.83 ± 1.27 |
| TVAE | 76.6 | 86.8 | 72.5 | 93.7 | 97.3 | 92.8 | 58.4 | 81.3 | 96.9 | 84.9 | 47.7 | 91.6 | 4.83 ± 1.53 |
| GReaT-distill | 76.5 | 87.6 | 72.4 | 93.6 | 98.0 | 92.7 | 58.4 | 77.6 | 96.4 | 84.9 | 53.0 | 90.4 | 5.17 ± 1.27 |
| GReaT | 75.8 | 87.6 | 72.1 | 93.5 | 98.2 | 93.0 | 58.5 | 80.1 | 97.9 | 85.8 | 53.4 | 91.4 | 4.08 ± 1.44 |
| TAPTAP-distill | 76.8 | **87.7** | 72.6 | **93.8** | **98.5** | 93.0 | 58.6 | **83.6** | **98.2** | 85.9 | **54.2** | 99.5 | 1.67 ± 0.49 |
| TAPTAP | 76.9 | **87.7** | 72.6 | **93.8** | 98.4 | **93.1** | **58.7** | **83.6** | 98.1 | **86.0** | 53.9 | 99.5 | 1.33 ± 0.49 |

Table 16: The experimental results in **low resource regime**. "+ Ori" means training with the original data. "+ Ori + Synthetic Data" means training with the original data plus the synthetic data. Below the backbone model is Transformer with piece-wise linear encoding.

| Metric ↑ | LO | AD | HE | CR | SI | BE | DI | CA | GE | ME | AG | DU | Avg. Rank |
|---|---|---|---|---|---|---|---|---|---|---|---|---|---|
| Transformer + Ori | 76.8 | 87.4 | 72.5 | 93.8 | 98.5 | 92.7 | 58.7 | 82.9 | 98.2 | 86.6 | 52.6 | 96.5 | - |
| *Transformer + Ori + Synthetic Data by Models* | | | | | | | | | | | | | |
| CTGAN | 74.7 | 87.2 | 71.5 | 84.8 | 97.8 | 92.7 | 58.5 | 81.5 | 96.3 | 72.1 | 51.6 | 71.7 | 5.79 ± 1.20 |
| CopulaGAN | 74.7 | 87.2 | 71.8 | **93.8** | 97.8 | 92.5 | 58.5 | 81.7 | 95.9 | 72.8 | 52.0 | 86.8 | 5.12 ± 1.38 |
| TVAE | 76.2 | 86.8 | **72.8** | 93.7 | 97.4 | 92.5 | 58.4 | 82.0 | 97.2 | 85.7 | 47.3 | 80.0 | 4.83 ± 1.90 |
| GReaT-distill | 76.1 | 87.5 | 72.0 | 93.6 | 98.3 | 92.6 | 58.4 | 77.9 | 96.6 | 86.2 | 52.4 | 79.0 | 5.00 ± 1.13 |
| GReaT | 74.5 | **87.6** | 72.1 | 93.6 | 98.4 | 92.7 | 58.5 | 80.5 | 98.1 | 86.4 | 53.3 | 80.3 | 3.92 ± 1.68 |
| TAPTAP-distill | 76.2 | **87.6** | 72.5 | **93.8** | **98.5** | 92.8 | 58.6 | **83.7** | **98.2** | **86.9** | 53.8 | **98.2** | 1.83 ± 0.58 |
| TAPTAP | **77.5** | 87.5 | 72.5 | **93.8** | **98.5** | **92.9** | **58.7** | **83.7** | **98.2** | 86.7 | 53.5 | 97.9 | 1.50 ± 0.67 |

Table 17: The experimental results in **missing value imputation**. "+ M-Ori" means training with the original data processed by the MCAR mechanism. "+ M-Ori + Synthetic Data" means training with the M-Ori data where the missing values are imputed by different models. Below the backbone model is LightGBM. ✗ denotes the method cannot run successfully on the dataset due to too many missing values.

| Metric ↑ | LO | AD | HE | CR | SI | BE | DI | CA | GE | ME | AG | DU | Avg. Rank |
|---|---|---|---|---|---|---|---|---|---|---|---|---|---|
| LightGBM + M-Ori | 73.3 | 86.2 | 71.3 | 93.7 | 97.0 | 91.1 | 57.4 | 68.0 | 93.1 | 66.6 | 44.2 | 83.5 | - |
| *LightGBM + M-Ori + Synthetic Data by Models* | | | | | | | | | | | | | |
| MIWAE | 71.0 | ✗ | 69.3 | ✗ | 96.8 | 90.3 | ✗ | 64.3 | 90.2 | 65.6 | 41.3 | 81.4 | 7.21 ± 0.94 |
| Sinkhorn | 73.8 | 84.5 | 69.4 | **93.7** | 96.8 | 89.2 | 57.0 | 66.5 | 93.3 | 67.9 | 50.8 | 81.2 | 6.00 ± 1.21 |
| MICE | 74.5 | 85.3 | 69.9 | 93.6 | 96.1 | 89.6 | 57.2 | 66.2 | 94.0 | 71.0 | 52.9 | 89.5 | 5.17 ± 1.47 |
| GAIN | **85.4** | 86.4 | **74.3** | 76.1 | 97.8 | 90.8 | **60.4** | 62.8 | 93.3 | 67.3 | 44.6 | 85.1 | 4.67 ± 2.64 |
| MissForest | 67.7 | 86.4 | 71.5 | **93.7** | 97.8 | 91.5 | 57.0 | 73.5 | 94.6 | 77.0 | 46.4 | 90.7 | 4.42 ± 1.62 |
| HyperImpute | 69.6 | **88.0** | 71.0 | 91.7 | 97.7 | 92.7 | ✗ | 80.8 | 96.3 | **79.8** | **57.3** | **92.2** | 3.46 ± 2.50 |
| TAPTAP-distill | 75.2 | 87.3 | 72.4 | **93.7** | **98.3** | **93.4** | 57.3 | 80.5 | 94.6 | 70.0 | 53.9 | 70.2 | 3.00 ± 1.91 |
| TAPTAP | 74.8 | 87.4 | 72.8 | **93.7** | 97.8 | 93.2 | 57.7 | **85.0** | **97.5** | 77.8 | 53.7 | 85.8 | 2.08 ± 0.90 |

Table 18: The experimental results in **missing value imputation**. "+ M-Ori" means training with the original data processed by the MCAR mechanism. "+ M-Ori + Synthetic Data" means training with the M-Ori data where the missing values are imputed by different models. Below the backbone model is Transformer. ✗ denotes the method cannot run successfully on the dataset due to too many missing values.

| Metric ↑ | LO | AD | HE | CR | SI | BE | DI | CA | GE | ME | AG | DU | Avg. Rank |
|---|---|---|---|---|---|---|---|---|---|---|---|---|---|
| Transformer + M-Ori | 73.4 | 85.5 | 71.2 | 93.6 | 96.7 | 90.6 | 57.4 | 63.8 | 93.5 | 71.0 | 41.9 | 67.1 | - |
| *Transformer + M-Ori + Synthetic Data by Models* | | | | | | | | | | | | | |
| MIWAE | 72.7 | ✗ | 68.7 | ✗ | 96.3 | 89.8 | ✗ | 61.0 | 90.9 | 68.8 | 42.8 | 74.0 | 7.46 ± 0.66 |
| Sinkhorn | 72.1 | 83.6 | 69.5 | 93.6 | 96.7 | 89.1 | 56.6 | 63.8 | 93.5 | 74.4 | 50.4 | 79.8 | 5.67 ± 1.56 |
| GAIN | **77.2** | 86.1 | 70.1 | 52.1 | 97.8 | 90.3 | 53.8 | 50.5 | 93.3 | 69.6 | 44.6 | 75.2 | 5.33 ± 2.19 |
| MICE | 73.0 | 84.7 | 69.9 | 93.6 | 95.5 | 89.0 | 57.6 | 64.2 | 93.8 | 74.1 | 52.1 | 76.7 | 5.25 ± 1.66 |
| MissForest | 73.0 | 83.9 | 70.7 | 92.8 | 97.3 | 91.6 | 57.3 | 74.6 | 94.7 | 78.7 | 46.3 | 82.8 | 4.00 ± 1.28 |
| HyperImpute | 75.3 | 86.7 | 69.8 | 83.6 | 97.2 | 92.8 | ✗ | 77.7 | 96.4 | 80.0 | **56.8** | **85.5** | 3.46 ± 2.15 |
| TAPTAP-distill | 74.6 | 86.9 | 72.3 | 93.6 | **98.0** | **93.3** | 57.2 | 79.0 | 94.6 | 75.0 | 53.2 | 68.4 | 2.92 ± 1.93 |
| TAPTAP | 73.1 | **87.0** | **72.7** | **93.7** | 97.5 | 93.2 | **57.8** | **83.6** | **97.6** | **82.5** | 52.4 | 78.6 | 1.92 ± 1.24 |

Table 19: The experimental results in **missing value imputation**. "+ M-Ori" means training with the original data processed by the MAR mechanism. "+ M-Ori + Synthetic Data" means training with the M-Ori data where the missing values are imputed by different models. Below the backbone model is LightGBM. ✗ denotes the method cannot run successfully on the dataset due to too many missing values.

| Metric ↑ | LO | AD | HE | CR | SI | BE | DI | CA | GE | ME | AG | DU | Avg. Rank |
|---|---|---|---|---|---|---|---|---|---|---|---|---|---|
| LightGBM + M-Ori | 77.1 | 86.9 | 72.1 | 93.7 | 97.3 | 91.8 | 58.5 | 80.1 | 93.7 | 50.5 | 52.0 | 82.9 | - |
| *LightGBM + M-Ori + Synthetic Data by Models* | | | | | | | | | | | | | |
| Sinkhorn | 77.1 | 86.3 | 69.9 | 93.8 | 97.3 | 91.2 | **58.7** | 78.6 | 93.5 | 49.8 | 50.3 | 61.3 | 6.08 ± 1.83 |
| MIWAE | 78.2 | 86.4 | 69.9 | ✗ | 97.3 | 91.6 | ✗ | 79.1 | 92.7 | 48.1 | 51.8 | 79.0 | 5.88 ± 1.68 |
| MICE | 77.1 | 86.7 | 70.4 | 93.8 | 96.6 | 91.2 | 58.3 | 79.4 | 93.7 | 62.8 | 52.4 | **93.2** | 5.33 ± 1.78 |
| GAIN | **78.8** | 87.0 | **75.4** | **97.0** | 97.2 | 90.0 | 52.8 | 69.7 | 88.3 | 45.9 | **55.0** | 74.0 | 4.92 ± 3.09 |
| MissForest | 77.0 | 87.1 | 68.7 | 95.9 | 97.9 | 92.3 | 58.5 | 80.1 | 93.8 | 84.2 | 51.9 | 59.0 | 4.67 ± 2.19 |
| HyperImpute | 77.1 | **90.0** | 70.7 | 94.5 | 97.9 | 92.3 | ✗ | 83.6 | 96.0 | **85.8** | 51.4 | 60.9 | 3.96 ± 2.38 |
| TAPTAP-distill | 77.3 | 87.6 | 72.5 | 93.8 | **98.3** | 92.8 | 58.5 | 81.0 | 93.9 | 73.1 | 53.0 | 83.3 | 2.83 ± 0.94 |
| TAPTAP | 77.3 | 87.5 | 72.6 | 93.8 | 98.0 | **93.1** | 58.6 | **83.9** | **97.0** | 77.7 | 53.1 | 79.1 | 2.33 ± 1.15 |

Table 20: The experimental results in **missing value imputation**. "+ M-Ori" means training with the original data processed by the MAR mechanism. "+ M-Ori + Synthetic Data" means training with the M-Ori data where the missing values are imputed by different models. Below the backbone model is MLP. ✗ denotes the method cannot run successfully on the dataset due to too many missing values.

| Metric ↑ | LO | AD | HE | CR | SI | BE | DI | CA | GE | ME | AG | DU | Avg. Rank |
|---|---|---|---|---|---|---|---|---|---|---|---|---|---|
| MLP + M-Ori | 77.3 | 86.2 | 72.0 | 93.6 | 96.4 | 91.7 | 58.4 | 76.9 | 93.5 | 53.0 | 52.1 | 76.9 | - |
| *MLP + M-Ori + Synthetic Data by Models* | | | | | | | | | | | | | |
| Sinkhorn | 77.3 | 85.3 | 69.9 | 93.7 | 96.3 | 91.5 | 58.4 | 76.5 | 93.2 | 45.5 | 51.4 | 76.7 | 6.29 ± 1.39 |
| MIWAE | 77.8 | 85.6 | 69.6 | ✗ | 96.7 | 91.5 | ✗ | 76.8 | 92.2 | 46.9 | 52.1 | 74.2 | 6.12 ± 1.71 |
| MICE | 77.2 | 86.0 | 70.1 | 93.7 | 95.7 | 91.0 | 58.3 | 77.1 | 93.3 | 61.5 | 51.9 | **97.2** | 5.50 ± 1.73 |
| GAIN | **78.1** | 85.7 | **75.3** | **97.0** | 95.5 | 87.6 | 55.3 | 61.6 | 78.3 | 46.9 | **54.5** | 70.1 | 5.25 ± 3.22 |
| MissForest | 77.3 | 86.1 | 70.4 | 96.6 | 97.3 | 92.2 | 58.4 | 78.4 | 93.6 | 83.6 | 52.3 | 75.9 | 4.04 ± 1.18 |
| HyperImpute | 77.3 | **88.6** | 70.8 | 94.4 | **98.1** | 92.6 | ✗ | 80.6 | 95.5 | **86.1** | 51.2 | 77.2 | 3.50 ± 2.42 |
| TAPTAP-distill | 77.3 | 87.0 | 72.5 | 93.8 | 97.6 | **93.4** | 58.5 | 79.2 | 93.6 | 73.8 | 52.5 | 84.2 | 2.83 ± 1.03 |
| TAPTAP | 77.3 | 87.0 | 73.1 | 93.8 | 97.4 | 93.2 | **58.6** | **81.2** | **97.0** | 77.4 | 52.6 | 81.7 | 2.46 ± 1.34 |

Table 21: The experimental results in **missing value imputation**. "+ M-Ori" means training with the original data processed by the MAR mechanism. "+ M-Ori + Synthetic Data" means training with the M-Ori data where the missing values are imputed by different models. Below the backbone model is Transformer. ✗ denotes the method cannot run successfully on the dataset due to too many missing values.

| Metric ↑ | LO | AD | HE | CR | SI | BE | DI | CA | GE | ME | AG | DU | Avg. Rank |
|---|---|---|---|---|---|---|---|---|---|---|---|---|---|
| Transformer + M-Ori | 76.0 | 86.2 | 72.2 | 93.7 | 97.0 | 91.8 | 58.4 | 77.9 | 93.6 | 54.6 | 51.9 | 72.2 | - |
| *Transformer + M-Ori + Synthetic Data by Models* | | | | | | | | | | | | | |
| MIWAE | 76.8 | 85.6 | 69.6 | ✗ | 96.9 | 91.6 | ✗ | 77.8 | 92.5 | 49.0 | 51.7 | 69.9 | 6.33 ± 1.42 |
| GAIN | 76.4 | 85.0 | **74.3** | **97.4** | 96.0 | 87.5 | 55.3 | 63.4 | 79.3 | 48.1 | **54.3** | 71.0 | 5.83 ± 3.01 |
| Sinkhorn | 76.6 | 85.1 | 69.8 | 93.7 | 96.6 | 91.6 | 58.4 | 77.1 | 93.3 | 48.3 | 51.7 | 74.9 | 5.67 ± 1.30 |
| MICE | 76.8 | 86.1 | 70.0 | 93.7 | 96.4 | 91.3 | 58.3 | 77.6 | 93.5 | 63.2 | 51.8 | **88.6** | 5.17 ± 1.70 |
| MissForest | 76.9 | 86.3 | 70.6 | 96.7 | 97.7 | 91.9 | 58.4 | 78.7 | 92.7 | 84.9 | 51.6 | 70.9 | 4.08 ± 1.78 |
| HyperImpute | 76.4 | **88.8** | 70.1 | 94.3 | 97.6 | 92.4 | ✗ | 81.0 | 95.8 | **87.7** | 50.4 | 76.0 | 3.96 ± 2.60 |
| TAPTAP-distill | **77.0** | 86.9 | 72.2 | 93.8 | **98.5** | 92.9 | 58.5 | 79.6 | 93.8 | 76.4 | 52.1 | 80.2 | 2.58 ± 1.16 |
| TAPTAP | 76.8 | 86.9 | 73.0 | 93.8 | 97.7 | **93.0** | **58.6** | **81.7** | **97.1** | 79.0 | 51.8 | 71.1 | 2.38 ± 1.33 |

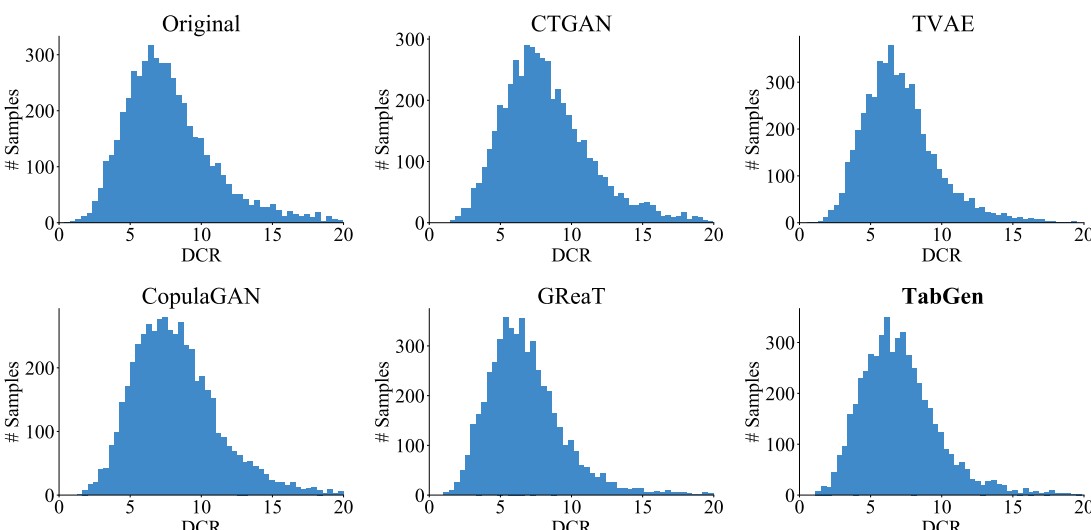

Figure 7: Distance to closest record (DCR) distribution of the California Housing dataset. "Original" denotes the DCR of the original test set with respect to the original train set. The experimental results illustrate that each method does not copy samples from the train set.

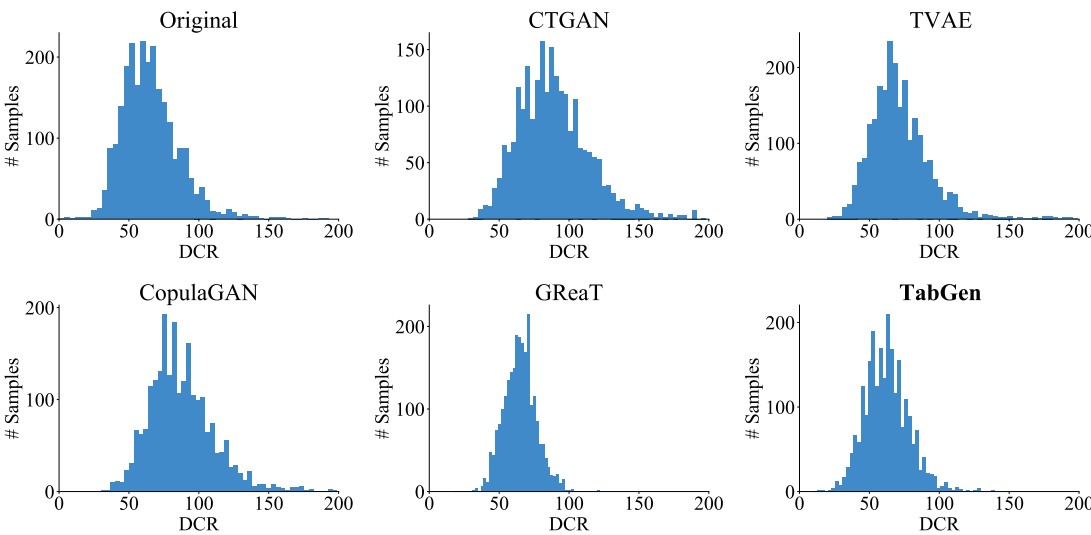

Figure 8: Distance to closest record (DCR) distribution of the HELOC dataset.