# OpenReview forum: "Generative Table Pre-training Empowers Models for Tabular Prediction"
_EMNLP/2023/Conference — EMNLP 2023 Main_

### Official Review · Reviewer_2Qac · 2023-08-01

**Soundness:** 3

**Excitement:**

4: Strong: This paper deepens the understanding of some phenomenon or lowers the barriers to an existing research direction.

**Paper Topic And Main Contributions:**

The target of this paper is to perform generative pre-training over a bunch of tabular datasets and then synthesize tabular data for downstream tasks. The synthetic data are then labeled by a classifier trained on the real data. Subsequently, the labeled synthetic data and the real data are combined to train a final classifier for the downstream task.

The authors emphasize the utility of this data synthesis approach in the following aspects:

- Privacy-preserving tabular prediction: replacing the real data with the synthesized data to build the predictive ML models.

- Few-shot tabular prediction: augmenting the real data to boost the predictive ML models in low-data scenarios.

- Missing data imputation: the generative pre-training model can help impute the missing values in the target table.

**Reasons To Accept:**

- The authors conducted a large-scale experiment over a bundle of 450 tabular datasets collected from OpenML and Kaggle. It would benefit the community if the authors could offer this data for future research in tabular learning.

- The TapTap method that has been proposed can be used for pre-training across various tables and fine-tuned to generate for targeting tabular data. The setting is reasonable and brings the promise of copying the success of generative pre-training on text to on tabular datasets.

**Reasons To Reject:**

- This paper sort of overclaims its contribution as "the first attempt to apply table pre-training to tabular prediction". There were many attempts in tabular pre-training to improve tabular prediction [1,2,3,4,5]. In my opinion, the major contribution of this paper lies in the cross-table generative pre-training and then generates synthetic tabular data for the target task to augment the downstream machine learning prediction models.

- The position of this paper seems to be suboptimal. While the paper emphasizes its capability in tabular prediction, the main strength of this paper, in my perspective, lies in its tabular data generation capability. From the observation made in Tables 3 and 5, the lift obtained by augmentation tabular data seems to be marginal over the other baselines. By contrast, in Table 2, it is witnessed that the proposed method reaches the lowest performance gap between the model trained on real data and trained on synthetic data, showing its power in generating synthetic data. It is suggested the author dives deeper into its generation capability and perform more in-depth of the fidelity, utility, and privacy of the synthetic data.


---

[1] Sercan O Arık and Tomas Pfister. Tabnet: Attentive interpretable tabular learning. In AAAI, volume 35, pages 6679–6687, 2021.

[2] Jinsung Yoon, Yao Zhang, James Jordon, and Mihaela van der Schaar. VIME: Extending the success of self-and semi-supervised learning to tabular domain. Advances in Neural Information Processing Systems, 33:11033–11043, 2020.

[3] Dara Bahri, Heinrich Jiang, Yi Tay, and Donald Metzler. SCARF: Self-supervised contrastive learning using random feature corruption. In International Conference on Learning Representations, 2022.

[4] Talip Ucar, Ehsan Hajiramezanali, and Lindsay Edwards. SubTab: Subsetting features of tabular data for self-supervised representation learning. Advances in Neural Information Processing Systems, 34, 2021.

[5] Wang Z, Sun J. Transtab: Learning transferable tabular transformers across tables. Advances in Neural Information Processing Systems, 2022, 35: 2902-2915.

**Reproducibility:**

3: Could reproduce the results with some difficulty. The settings of parameters are underspecified or subjectively determined; the training/evaluation data are not widely available.

**Reviewer Confidence:**

5: Positive that my evaluation is correct. I read the paper very carefully and I am very familiar with related work.

---

> ### Author Rebuttal · Authors · 2023-08-27
>
> Thank you for your constructive review!
>
> > This paper sort of overclaims its contribution as "the first attempt to apply table pre-training to tabular prediction". There were many attempts in tabular pre-training to improve tabular prediction [1,2,3,4,5]. In my opinion, the major contribution of this paper lies in the cross-table generative pre-training and then generates synthetic tabular data for the target task to augment the downstream machine learning prediction models.
>
> We sincerely appreciate the reviewer’s thoughtful comments and critique. We agree that there were other previous studies that performed tabular pre-training. The major difference between TAPTAP and previous studies on tabular pre-training is that, TAPTAP performs cross-table pre-training using a large number of tables on language models to leverage the knowledge embedded in language models, and previous works (such as TabNet, VIME, and SCARF) usually perform single-table pre-training (or few tables with lots of overlapped columns, such as TransTab) on models specifically designed for tabular data. There are other studies that perform cross-table pre-training on language models (such as TaPaS, TaBERT, and TaPEx) for joint reasoning over text and tables (such as the TableQA task), yet TAPTAP is the first attempt to apply cross-table pre-training on language models for the tabular prediction task.
>
> We will refine the claim for better clarity in the revised version according to your suggestions. We will also include an extended discussion on the differences between TAPTAP and the tabular pre-training studies referenced by the reviewer in the revised version of our paper.
>
> > The position of this paper seems to be suboptimal. While the paper emphasizes its capability in tabular prediction, the main strength of this paper, in my perspective, lies in its tabular data generation capability. From the observation made in Tables 3 and 5, the lift obtained by augmentation tabular data seems to be marginal over the other baselines. By contrast, in Table 2, it is witnessed that the proposed method reaches the lowest performance gap between the model trained on real data and trained on synthetic data, showing its power in generating synthetic data. It is suggested the author dives deeper into its generation capability and perform more in-depth of the fidelity, utility, and privacy of the synthetic data.
>
> We greatly appreciate your keen observations about the positioning of our paper in the context of its strength in tabular data generation. We agree that the main strength of this paper is the tabular data generation capability. We determine the current positioning of this paper because a key purpose of tabular data modeling is to boost the performance of tabular prediction, and the merits of tabular data generation are substantiated by its role in improving the performance of tabular prediction in different challenging scenarios (such as privacy protection, missing value imputation, etc). We will follow your suggestions and include these discussions in the revised version.
>
> While the marginal lift over baselines in the data augmentation experiments may not seem impressive, we note that improving performance with synthetic data is a challenging problem. Despite this difficulty, TAPTAP still manages to outperform other state-of-the-art techniques, demonstrating its capabilities in synthesizing useful tabular data.

---

### Official Review · Reviewer_YvRN · 2023-08-03

**Typos Grammar Style And Presentation Improvements:** 1. Some details are unclear in the ma…
**Soundness:** 4

**Excitement:**

4: Strong: This paper deepens the understanding of some phenomenon or lowers the barriers to an existing research direction.

**Paper Topic And Main Contributions:**

Table pretraining has been well-explored to solve the joint reasoning of tables and text, but its effectiveness for tabular prediction tasks has not been verified. This paper presents the first successful trial of table pretraining for tabular prediction by proposing a TAPTAP framework to enhance downstream tabular prediction tasks by synthesizing high-quality table examples. They collected a large-scale pretraining corpus to enable the pretraining, and systematically evaluated the framework on 12 datasets across four downstream task scenarios: privacy protection, low-resource regime, missing value imputation and imbalanced classification. Experimental results have shown the benefits of the generative pretrained model for different backbone models (e.g., GBDT models or Transformer which are trained for downstream tasks).

**Questions For The Authors:**

A. See the above Reason to Reject 1.

**Reasons To Accept:**

1. The proposed data generator model is novel and has good generalizability to various tabular prediction tasks.
2. The design of the proposed method, including input encoding and data augmentation, provides useful empirical insights that may benefit the research in this field.
3. The experiments are thorough, showing the effectiveness of the proposed method in a wide range of scenarios.

**Reasons To Reject:**

1. It lacks a qualitative analysis of the quality of the synthesized table examples. The tabular prediction tasks require learning a distribution across the feature values and labels, but how can you guarantee that the synthesized feature values in the augmented tables are beneficial rather than noisy for the learning of downstream models?

**Reproducibility:**

5: Could easily reproduce the results.

**Reviewer Confidence:**

4: Quite sure. I tried to check the important points carefully. It's unlikely, though conceivable, that I missed something that should affect my ratings.

---

> ### Author Rebuttal · Authors · 2023-08-27
>
> Thank you for your insightful review!
>
> > It lacks a qualitative analysis of the quality of the synthesized table examples. The tabular prediction tasks require learning a distribution across the feature values and labels, but how can you guarantee that the synthesized feature values in the augmented tables are beneficial rather than noisy for the learning of downstream models?
>
> We measure the quality of the synthesized samples by their performance in the application scenarios. There is substantial evidence in our experiments that the synthesized feature values are not just noise but substantive contributors to the learning performance of downstream models. For example, on half of the privacy protection datasets, models trained with synthetic data delivered performance similar with those trained using the original data. Furthermore, when additional synthetic features (or samples) generated by TAPTAP are integrated into the training data, the machine learning models show great improvements, outperforming those trained with original datasets in the missing value and imbalanced classification scenarios.
>
> > Some details are unclear in the main body: the exact base model for TAPTAP, the description of TAPTAP-distill, etc.
>
> Thank you for pointing it out. TAPTAP uses the original GPT2 [1] with 355M parameters, while TAPTAP-distill uses the distilled version of GPT2 [2] with 82M parameters. We will include this information in the experimental setup section in the revised version.
>
> [1] Radford A, Wu J, Child R, et al. Language models are unsupervised multitask learners[J]. OpenAI blog, 2019, 1(8): 9.
>
> [2] Sanh V, Debut L, Chaumond J, et al. Distilbert, a distilled version of BERT: smaller, faster, cheaper and lighter[J/OL]. CoRR, 2019, abs/1910.01108. http://arxiv.org/abs/1910.01108.

---

### Official Review · Reviewer_PTdH · 2023-08-04

**Soundness:** 4

**Excitement:**

4: Strong: This paper deepens the understanding of some phenomenon or lowers the barriers to an existing research direction.

**Paper Topic And Main Contributions:**

The authors proposed TAPTAP(Table Pretraining for Tabular Prediction),  a method that aims to improve the quality of the training data for tabular prediction. After pretraining language model in the public tabular datasets via static prompt (e.g., <column> is <value>), fine-tuning LM, and data sampling, the resulting synthetic data can help resolve four common challenges: (1) Privacy Protection, (2) Low Resource Regime, (3) Missing Value Imputation, and (4) Imbalanced Classification. The experimental results show that the proposed TAPTAP and TAPTAP-distill significantly improve the performance of tasks (1)-(4) over 12 publicly available datasets.

**Questions For The Authors:**

- Question A: In the experiment of missing value prediction (Table 4, Table 17-21), some methods failed due to massive missing values. However, TAPTAP seems robust due to the permutation function (L306-L323). Does the permutation function somehow a data augmentation module, or what it does is just shuffling the data?
- Question B: In section 3.5, how is the percentage of data prompt decided? Is it data-specific, or it depends on which task is solving? For example, use (1) Feature name as prompt for data privacy tasks.


**Reasons To Accept:**

1. Four challenging tasks, especially Privacy Protection and Missing Data Imputation, are crucial in medical data engineering as patient data cannot be shared across hospitals. In addition, the legacy software in the hospital often makes ETL a barrier to introducing grounded AI in this field. Hence, a method for generating high-quality training data like this work is worth exploring.
2. The authors provide a comprehensive report and implementation details (section 3.3) on applying the proposed method for resolving tasks (1)-(4), including using different backbone models (i.e., MLP, LightBGM, and Transformer) on 12 datasets that fairly report the pros and cons of baselines and TAPTAP. The reproducibility of this work is good.

**Reasons To Reject:**

1. Though mentioned in L86-L91, linking the purpose of tabular prediction and the experiments takes a lot of work in the first pass due to the intense information in this paper. Not until L212-L214, it's hard to catch the main contribution of TAPTAP is to improve the quality of training data, which is helpful for tasks (1)-(4).


**Reproducibility:**

5: Could easily reproduce the results.

**Reviewer Confidence:**

4: Quite sure. I tried to check the important points carefully. It's unlikely, though conceivable, that I missed something that should affect my ratings.

---

> ### Author Rebuttal · Authors · 2023-08-27
>
> Thank you for your constructive review!
>
> > Question A: In the experiment of missing value prediction (Table 4, Table 17-21), some methods failed due to massive missing values. However, TAPTAP seems robust due to the permutation function (L306-L323). Does the permutation function somehow a data augmentation module, or what it does is just shuffling the data?
>
> The permutation function serves as a kind of data augmentation in TAPTAP. Each time a training sample is feed into the model, the order of its features is randomly shuffled. Through this process, the permutation function reconstructs the feature order independence. Consequently, TAPTAP can flexibly sample from any provided feature-value pairs in a robust way, and thus perform well in the missing value imputation scenario.
>
>
> > Question B: In section 3.5, how is the percentage of data prompt decided? Is it data-specific, or it depends on which task is solving? For example, use (1) Feature name as prompt for data privacy tasks.
>
> The selection of data prompting strategy depends on the characteristics of the task at hand (i.e., the quantity of feature values that need to be sampled), and hence is task-dependent rather than data-specific. For example, tasks such as privacy protection and low resource regimes necessitate the sampling of all feature values, leading to the adoption of the 'feature-name-as-prompt' strategy. Conversely, the task of missing value imputation requires the sampling of absent feature values, given the presence of known feature values. As a result, it employs the 'multiple-feature-value-pairs-as-prompt' strategy.

---

### Meta-Review · Area_Chair_kudU · 2023-09-16

**Recommendation:** 3

**Metareview:**

The paper titled "TAPTAP: Generating High-Quality Training Data for Tabular Prediction" presents a method called TAPTAP for improving the quality of training data for tabular prediction tasks. The proposed method involves pretraining a language model on public tabular datasets using static prompts, fine-tuning the language model, and data sampling. The resulting synthetic data helps address four common challenges in tabular prediction: privacy protection, low resource regime, missing value imputation, and imbalanced classification. The paper presents experimental results on 12 publicly available datasets that demonstrate the effectiveness of the proposed method in improving the performance of these tasks. Overall, the reviewers think that the work has done well in this task, and also has some concerns about the writing, claim. For example, the position of this paper has been questioned by one reviewer, some contribution might be overclaimed, and more experimental explanations and discussions are needed, too.

---

### Decision · Program_Chairs · 2023-10-07

**Decision:**

Accept-Main

**Comment:**

The paper titled "TAPTAP: Generating High-Quality Training Data for Tabular Prediction" presents a method called TAPTAP for improving the quality of training data for tabular prediction tasks. The proposed method involves pretraining a language model on public tabular datasets using static prompts, fine-tuning the language model, and data sampling. The resulting synthetic data helps address four common challenges in tabular prediction: privacy protection, low resource regime, missing value imputation, and imbalanced classification. The paper presents experimental results on 12 publicly available datasets that demonstrate the effectiveness of the proposed method in improving the performance of these tasks. Overall, the reviewers think that the work has done well in this task, and also has some concerns about the writing, claim. For example, the position of this paper has been questioned by one reviewer, some contribution might be overclaimed, and more experimental explanations and discussions are needed, too.